# BAG-OF-VECTORS AUTOENCODERS FOR UNSUPERVISED CONDITIONAL TEXT GENERATION

## ABSTRACT

Text autoencoders are often used for unsupervised conditional text generation by applying mappings in the latent space to change attributes to the desired values. Recently, Mai et al. (2020) proposed Emb2Emb, a method to *learn* these mappings in the embedding space of an autoencoder. However, their method is restricted to autoencoders with a single-vector embedding, which limits how much information can be retained. We address this issue by extending their method to *Bag-of-Vectors Autoencoders* (BoV-AEs), which encode the text into a variable-size bag of vectors that grows with the size of the text, as in attention-based models. This allows to encode and reconstruct much longer texts than standard autoencoders. Analogous to conventional autoencoders, we propose regularization techniques that facilitate learning meaningful operations in the latent space. Finally, we adapt Emb2Emb for a training scheme that learns to map an input bag to an output bag, including a novel loss function and neural architecture. Our empirical evaluations on unsupervised sentiment transfer and sentence summarization show that our method performs substantially better than a standard autoencoder.

## 1 INTRODUCTION

In conditional text generation, we would like to produce an output text given an input text. Hence, parallel input-output pairs are required to train a good supervised machine learning model on this type of task. Large-scale pretraining (Peters et al., 2018; Devlin et al., 2019; Lewis et al., 2020) can alleviate the necessity for training examples to some extent, but even this requires a substantial number of annotations (Yogatama et al., 2019). This is an expensive process and can introduce unwanted artifacts itself, which are henceforth learned by the model (Gururangan et al., 2018). For these reasons, there is substantial interest in unsupervised solutions. *Text autoencoders* (AEs) don't require labeled data for training, and are therefore a popular model for unsupervised approaches to many tasks, such as machine translation (Artetxe et al., 2018), sentence compression (Févry & Phang, 2018) and sentiment transfer (Shen et al., 2017). The classical text AE (Bowman et al., 2016) embeds the input text into a single fixed-size vector via the encoder, and then tries to reconstruct the input text from the single vector via the decoder. Single-vector embeddings are very useful, because they allow to perform conditional text generation through simple mappings in the embedding space, e.g. by adding a constant offset vector to change attributes such as sentiment (Shen et al., 2020). Recently, Mai et al. (2020) proposed Emb2Emb, a method that can *learn* these mappings directly in the embedding space of any pretrained single-vector AE. This is a powerful framework, because the AE can then be pretrained on virtually infinite amounts of unlabeled data before applying it to any downstream application. This concept, *transfer learning*, is arguably one of the most important drivers of progress in machine learning in the recent decade: These so-called *Foundation Models* (Bommasani et al., 2021) have revolutionized natural language understanding (e.g, *BERT* (Devlin et al., 2019)) and computer vision (e.g, *DALL-E* (Ramesh et al., 2021)), among others. Since Emb2Emb was designed to work with any pretrained AE, it was an important step towards their *scalability*.

However, as Bommasani et al. (2021) point out, another crucial model property is *expressivity*, the ability to represent the data distribution it is trained on. In this regard, single-vector representations are fundamentally limited; they act as a bottleneck, causing the model to increasingly struggle to encode longer text (Bahdanau et al., 2015). In this paper, we extent Emb2Emb from single-vector bottleneck AEs to *Bag-of-Vector Autoencoders* (BoV-AEs), which encode text into a variable-size representation where the number of vectors grows with the length of the text. This gives BoV-

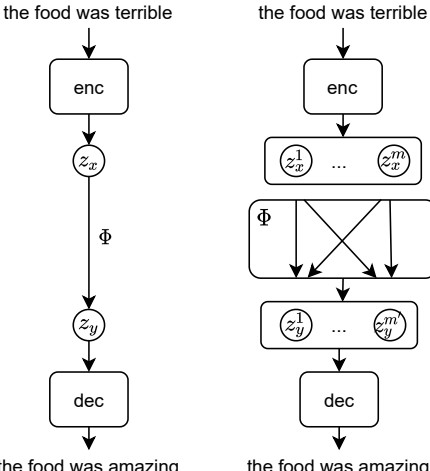

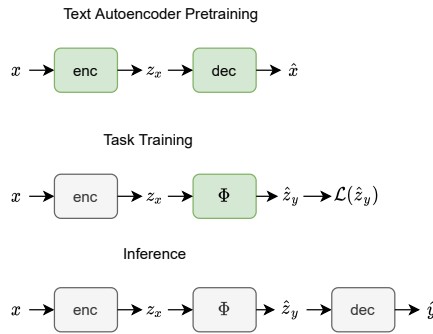

Figure 2: High-level view of the Emb2Emb framework. *Text Autoencoder Pretraining*: An autoencoder is trained on an unlabeled corpus, i.e., the encoder enc transforms an input text $x$ into a continuous embedding $\mathbf{z}_x$, which is in turn used by the decoder dec to predict a reconstruction $\hat{x}$ of the input sentence. *Task Training*: The encoder is frozen (grey), and a mapping $\Phi$ is trained (green) on input embeddings $\mathbf{z}_x$ to output predictions $\hat{\mathbf{z}}_y$ such that it minimize some loss $\mathcal{L}(\hat{\mathbf{z}}_y)$. *Inference*: To obtain textual predictions $\hat{y}$, the encoder is composed with $\Phi$ and the decoder.

Figure 1: *Left*: In the standard setup, the representation consists of a single vector, requiring a simple vector-to-vector mapping to do operations in the vector space. *Right*: In BoV-AE, the representation consists of a variable-size bag of vectors, requiring a more complex mapping from one bag to another bag.

AEs the same kind of representations as attention-based models. But this added expressivity comes with additional challenges, as illustrated in Figure 1. In the single-vector case, an operation $\Phi$ in the vector space consists of a simple vector-to-vector mapping. But with BoV-AEs, $\Phi$ needs to map a bag of vectors onto another bag of vectors, which requires more complicated operations. In this paper, we demonstrate how such a mapping can be learned in the context of the Emb2Emb framework by making the following novel contributions: **(i)** We propose a regularization scheme for BoV-AEs, **(ii)** a neural mapping architecture $\Phi$ for Emb2Emb, and **(iii)** a suitable training loss.

Empirically, we show on two unsupervised conditional text generation tasks, sentiment transfer (Shen et al., 2017) and sentence summarization (Rush et al., 2015), that BoV-AEs perform substantially better than standard AEs if the text is too long to be captured by one vector alone. Our ablation studies confirm that our technical contributions are crucial for this success.

In the following section, we review the Emb2Emb framework, before we introduce BoV-AE (Section 3) and its integration within Emb2Emb (Section 4).

## 2 BACKGROUND: Emb2Emb

*Embedding-to-Embedding* (Emb2Emb) was introduced by Mai et al. (2020) as both a supervised and an unsupervised framework for conditional text generation. The core idea is to disentangle the transition from the discrete text space to a continuous latent space from the specific task, allowing for larger-scale pretraining with unlabeled data.

The workflow of the framework is depicted in Figure 2. First, a text AE $\mathcal{A} = \text{dec} \circ \text{enc}$ is trained to map an input sentence from the discrete text space $\mathcal{X}$ to an embedding space $\mathcal{Z}$ via the encoder $\text{enc} : \mathcal{X} \to \mathcal{Z}$, and back to $\mathcal{X}$ via a decoder $\text{dec} : \mathcal{Z} \to \mathcal{X}$, such that $\mathcal{A}(x) = x$, typically trained via negative log-likelihood, $\mathcal{L}_{rec} = \text{NLL}(\mathcal{A}(x), x)$. In contrast to other methods, $\mathcal{A}$ can in principle be any AE, opening the possibility for large-scale AE pretraining with unlabeled data. Second, task-specific training is performed only in the embedding space $\mathcal{Z}$ of the AE. To this end, the encoder is frozen, and a new mapping layer $\Phi : \mathcal{Z} \to \mathcal{Z}$ is introduced, which is trained to transform the embedding of the input $\mathbf{z}_x$ into the embedding of the predicted output $\hat{\mathbf{z}}_y$. The concrete loss $\mathcal{L}(\hat{\mathbf{z}}_y)$ depends on the type of task. In the supervised case, the true output is also encoded into space

$\mathcal{Z}$, and the distance between the true embedding and the predicted embedding is minimized. In the unsupervised case, the loss needs to be defined for the specific task at hand. For example, for sentiment transfer, where the goal is to transform a negative review into a positive review while retaining as much of the input as possible, Mai et al. (2020) compose the loss as a combination of two loss terms[1], $\mathcal{L}(\hat{\mathbf{z}}_y) = \mathcal{L}_{sim}(\mathbf{z}_x, \hat{\mathbf{z}}_y) + \lambda_{sty}\mathcal{L}_{sty}(\hat{\mathbf{z}}_y)$. $\mathcal{L}_{sty}$ encourages $\hat{\mathbf{z}}_y$ to be classified as a positive review according to a separately trained sentiment classifier. $\mathcal{L}_{sim}$ encourages the output to be close to the input in embedding space, e.g. via euclidean distance. $\lambda_{sty}$ is a hyperparameter that controls the importance of changing the sentiment of the predicted output.

A main question in Emb2Emb is how to choose the embedding space $\mathcal{Z}$. Mai et al. (2020) use a single continuous vector to encode all the information of the input, i.e. $\mathcal{Z} = \mathbb{R}^d$. This choice simplifies the mapping $\Phi$ to an MLP and the training loss to vector space distances, which is relatively easy to train. On the other hand, it limits the model in fundamental ways: The representation is *fixed-sized*, i.e., the representation cannot grow in size. Sequence-to-sequence models with a fixed-size bottleneck struggle to encode long text sequences (Bahdanau et al., 2015), which is a key reason why attention-based models are now standard practice in sequence-to-sequence models. Hence, it would be desirable to adapt Emb2Emb in such a way that $\mathcal{Z}$ contains *variable-sized* embeddings instead.

## 3 BAG-OF-VECTORS AUTOENCODER

We propose *Bag-of-Vectors Autoencoders* (BoV-AEs) to be used with Emb2Emb. Following the naming convention by Henderson (2020), we refer to a bag of vectors as a (multi)-set of vectors that (i) can grow arbitrarily large, and (ii) where the elements are not ordered (a basic property of sets). A type of BoV representation that is used very commonly is found in Transformer (Vaswani et al., 2017) encoder-decoder models, where there is one vector to represent each token of the input text, and the order of the vectors does not matter when the decoder accesses the output of the encoder. In this work, we also rely on Transformer models as the backbone of our encoders and decoders. However, in principle, any encoder and decoder can be used, as long as the encoder produces a bag as output and the decoder takes a bag as input. Formally, $\mathcal{Z} = (\mathbb{R}^d)^+$, so the encoder produces a bag-of-vectors $\mathbb{X} = \{\mathbf{z}_1, ..., \mathbf{z}_n\} := \text{enc}(x)$, where $n$ is the number of vectors in the input bag.

### 3.1 REGULARIZATION

The fact that we use a BoV-based AE presents a major challenge: AEs have to be regularized to prevent them from learning a simple identity mapping where the input is merely copied to the output, which does not result in a meaningful embedding space. In fixed-size embeddings, this is for example achieved through under-completeness (choosing a latent dimension that is smaller than the input dimension) or through injection of noise, either at the input or in the embedding space. While there exists a lot of research on regularizing fixed-sized AEs, it is not clear how to achieve the same goal in a BoV-AE. Here, regularizing the capacity of each vector is not enough. As long as each vector can store a (constant) positive amount of information, a bag of unlimited size can still store infinite information. However, it is not clear to what extent the size of the bag needs to be restricted. By default, a standard Transformer model produces as many vectors as there are input tokens, but this is likely too many, as it makes copying from the input to the output trivial. Hence, we want the encoder to output fewer vectors. In the following we explain how this is achieved in BoV-AEs.

**L0Drop** Ideally, we want the model to decide for itself on a per-example basis which vectors it needs to retain for reconstruction. To this end, we adopt *L0Drop*, a differentiable approximation to L0 regularization, which was originally developed by Zhang et al. (2021) for the purpose of speeding up a model through sparsification. The model computes scalar gates $g_i = g(\mathbf{z}_i) \in [0, 1]$ (which can be exactly zero or one) for each encoder output. After the gates are computed, we multiply them with their corresponding vector. Vectors whose gates are near zero (i.e., smaller than some $\epsilon > 0$) are removed from the bag entirely. An additional loss term, $\mathcal{L}_{L_0}(\mathbb{X}) = \lambda_{L0} \sum_i^n g_i$ encourages the model to close as many gates as possible, where the hyperparameter $\lambda_{L0}$ controls the sparsity rate *implicitly*. However, in initial experiments, we found $\lambda_{L0}$ difficult to tune, as it is very sensitive with respect to other hyperparameters. We instead employ a modified loss that seeks to *explicitly* match

---

[1]Their total loss includes an adversarial component that encourages the outputs of the mapping to stay on the latent space manifold. We leave adaptation of this component to the BoV scenario for future work.

a certain target ratio $r$ of open gates. Similar to the *free-bits* objective that is used to prevent the posterior collapse problem in VAEs (Kingma et al., 2016), the objective becomes

$$\mathcal{L}_{L_0}(\mathbb{X}) = \lambda_{L0} \max(r, \tfrac{1}{n} \sum_i^n g_i). \tag{1}$$

By setting $\lambda_{L0}$ to a large enough value (empirically, $\lambda_{L0} = 10$), we find that this objective reaches the target ratio $r$ reliably for different $r$ while at the same time reducing the reconstruction loss. This allows to compare different strengths of regularization while reducing the tuning effort substantially.

## 4 EMB2EMB WITH BOV-AES

In the following we describe how to adapt the Emb2Emb model to BoV-AEs, i.e., how to generate an output bag $\hat{\mathbb{X}} = \{\hat{\mathbf{z}}_1, \ldots, \hat{\mathbf{z}}_n\}$ given an input bag $\mathbb{X}$ through the mapping $\Phi(\mathbb{X})$, and how to choose the loss function $\mathcal{L}(\hat{\mathbb{X}}, \mathbb{X})$. For example, in the case of style transfer, we want $\hat{\mathbb{X}}$ to be similar to $\mathbb{X}$.

### 4.1 MAPPING $\Phi$

In contrast to Mai et al. (2020), who use a single-vector embedding and hence $\Phi$ can be as simple as an MLP, in our work, $\Phi$ must be capable of producing a bag of vectors. The straight-forward choice for $\Phi$ is a Transformer decoder that uses cross-attention on the input BoV, and generates vectors autoregressively one at a time, formally $\hat{\mathbf{z}} = \mathrm{Transformer}(\mathbf{z}_s, \hat{\mathbf{z}}_1, \ldots, \hat{\mathbf{z}}_{t-1}, \mathbb{X}), t \geq 1$, where $\mathbf{z}_s$ is the embedding of some starting symbol. Depending on the difficulty of the task, this mapping may be sufficiently powerful, in particular for short texts (i.e. bags with few vectors). For longer texts, however, learning to map from one bag to another is difficult, and may hence require the integration of inductive biases.

Based on the assumption that the output should be close to the input in embedding space, Mai et al. (2020) propose *OffsetNet* for the single vector case, which computes an offset vector to be added to the input. With a similar motivation, we propose a variant of pointer-generator networks (See et al., 2017), which allows the model to choose between copying an input vector and generating a new one. Instead of just copying, however, our model ($\mathrm{Transformer}++$) allows to compute an offset vector to be added to the copied vector, analogous to Mai et al. (2020). Formally, at each timestep $t$,

$$\hat{\mathbf{z}}_t = (1 - p_{gen})(\mathbf{z}_{copy} + \mathbf{z}_{\mathrm{offset}}) + p_{gen}\mathbf{z}_t', \tag{2}$$

where $\mathbf{z}_t' = \mathrm{Transformer}(\mathbf{z}_s, ..., \hat{\mathbf{z}}_{t-1}, \mathbb{X})$. Intuitively, by controlling $p_{gen} \in (0, 1)$, the model makes the (soft) decision to either copy a vector from the input and add an offset, or to generate a completely new vector. Here, $p_{gen}$ is a function of $\mathbf{z}_t'$ and the starting symbol which we treat as a context vector, $p_{gen} = \sigma(\mathbf{W}[\mathbf{z}_s; \mathbf{z}_t'])$. Similarly, $\mathbf{z}_{\mathrm{offset}}$ is a one-layer MLP with $[\mathbf{z}_t'; \mathbf{z}_{copy}]$ as input. $\mathbf{z}_{copy}$ is determined through an attention function:

$$\mathbf{z}_{copy} = \sum_{i=1}^{|\mathbb{X}|} \alpha_i \mathbf{z}_i, \quad \mathbf{K} = \mathbf{W}_{cpy}\mathbf{X}, \quad \alpha_i = \mathrm{softmax}(\mathbf{z}_s^T \mathbf{K})_i, \quad (\mathbf{X})_i := \mathbf{z}_i \tag{3}$$

where $\mathbf{W}_{cpy}$ is a learnable weight matrix. We refer to this model as $\mathrm{Transformer}++$.

### 4.2 GENERATING VARIABLE SIZED BAGS

The output bag is generated in an autoregressive manner. In the unsupervised case, it is not always clear how many vectors the bag should contain. However, due to the unsupervised nature, all information needed for computing the (task-dependent) training loss $\mathcal{L}(\hat{\mathbb{X}}, \mathbb{X})$ are also available at inference time. In this case, we can first generate some fixed maximum number $N$ of vectors autoregressively, and then determine the optimal bag by computing the minimal (inference-time) loss value, $\mathbb{X}^* = \min_{l=1,\ldots,N} \mathcal{L}(\hat{\mathbb{X}}_{1:l}, \mathbb{X})$. This can be valuable for tasks where we do not have a good prior on the size of the target bag. During training, we minimize the loss locally at every step. But we don't necessarily care about the loss at very small or big bags, so we might want to weight the steps as $\mathcal{L}^{\mathrm{total}}(\hat{\mathbb{X}}, \mathbb{X}) = \sum_{l=1}^{N} \mathbf{w}_l \mathcal{L}(\hat{\mathbb{X}}_{1:l}, \mathbb{X})$. Here, $\mathbf{w} \in \mathbb{R}_+^N$ could be any weighting, but it is more beneficial for training to only backpropagate from bag sizes that we expect to be close to the optimal

output bag size. For instance, in style transfer, the output typically has about the same length as the input. Hence, for an input size of length $n$, a useful weighting could be

$$\mathbf{w}_l = \begin{cases} 1 & n - k \leq l \leq n + k \\ 0 & \text{otherwise} \end{cases}, \tag{4}$$

in which $k$ denotes the size of a window around the input bag size.

### 4.3 Aligning Two Bags of Vectors

As described in Section 2, unsupervised sentiment transfer involves two loss terms, $\mathcal{L}_{sty}$ and $\mathcal{L}_{sim}$. In order to adapt $\mathcal{L}_{sty}$ from the single vector case to the BoV case, we can simply switch from an MLP classifier to a Transformer-based classifier. For $\mathcal{L}_{sim}$, however, we need to switch to a loss function that is defined on sets. While there are well-known losses for the single-vector case, in NLP set-level loss functions are not well-studied.

Here, we propose a novel variant of the *Hausdorff* distance. This distance is commonly used in vision applications: as a performance evaluation metric in e.g. medical image segmentation (Taha & Hanbury, 2015; Aydin et al., 2020), or in vision systems as a way to compare images (Huttenlocher et al., 1992; Takács, 1998; Lin et al., 2003; Lu et al., 2001). More recently, variants (different from ours) of the Hausdorff distance have also been used as loss functions to train neural networks (Fan et al., 2017; Ribera et al., 2019; Zhao et al., 2021). In NLP, its use is very rare (Nutanong et al., 2016; Chen, 2019; Kuo et al., 2020). To the best of our knowledge, our paper is the first to present a novel, fully differentiable variant of the Hausdorff distance as a loss for language learning.

The Hausdorff distance is a method for aligning two sets. Given two sets $\mathbb{X}$ and $\hat{\mathbb{X}}$, their Hausdorff distance H is defined as

$$\text{H}(\mathbb{X}, \hat{\mathbb{X}}) = \frac{1}{2} \text{align}(\mathbb{X}, \hat{\mathbb{X}}) + \frac{1}{2} \text{align}(\hat{\mathbb{X}}, \mathbb{X}), \quad \text{align}(\mathbb{X}, \hat{\mathbb{X}}) = \max_{x \in \mathbb{X}} \min_{y \in \hat{\mathbb{X}}} \text{d}(x, y) \tag{5}$$

Intuitively, two sets are close if each point in either set has a counterpart in the other set that is close to it according to some distance metric $d$. We choose $d$ to be the euclidean distance, but in principle any differentiable distance metric could be used (e.g. cosine distance). However, the vanilla Hausdorff distance is very prone to outliers, and therefore often reduced to the *average Hausdorff distance* (Dubuisson & Jain, 1994), where

$$\text{align}(\mathbb{X}, \hat{\mathbb{X}}) = \frac{1}{|\mathbb{X}|} \sum_{x \in \mathbb{X}} \min_{y \in \hat{\mathbb{X}}} d(x, y). \tag{6}$$

The average Hausdorff function is step-wise smooth and differentiable. Empirically, however, we find step-wise smoothness to be insufficient for the best training outcome. Therefore, we propose a fully differentiable version of the Hausdorff distance by replacing the $\min$ operation with $\text{softmin}$ like follows:

$$\text{align}(\mathbb{X}, \hat{\mathbb{X}}) = \frac{1}{|\mathbb{X}|} \sum_{x \in \mathbb{X}} \sum_{y \in \hat{\mathbb{X}}} \left( \frac{e^{(-d(x,y))}}{\sum\limits_{y' \in \hat{\mathbb{X}}} e^{(-d(x,y'))}} \cdot d(x, y) \right). \tag{7}$$

This variant is reminiscent of the attention mechanism Bahdanau et al. (2015) in the sense that a weighted average is computed, which has been very successful at smoothly approximating discrete decisions, e.g., read and write operations in the Differentiable Neural Computer (Graves et al., 2016) among many others.

## 5 Experiments

Our experiments are designed to test the following two hypotheses. **H1**: If the input text is too long to be encoded into a fixed-size single vector representation, BoV-AE-based Emb2Emb provides a substantial advantage over the fixed-sized model. **H2**: Our technical contributions, namely L0Drop regularization, the training loss, and the mapping architecture, are necessary for BoV-AE's success.

We evaluate our model on two unsupervised conditional text generation tasks: In Section 5.1, we show that **H1** holds even when the single-vector dimensionality is large ($d$=512). To this end, we

create a new sentiment transfer dataset, Yelp-Reviews, whose inputs are relatively long. However, training on this dataset is computationally very demanding[2]. Therefore, we turn to shorter text datasets to test hypothesis **H2**. Concretely, we test on a sentiment transfer dataset of short texts (Section 5.2), and a sentence summarization dataset of medium length texts (Section 5.3).

**Evaluation metrics:** In sentiment transfer, the goal is to rewrite a negative review as a positive review while keeping as much of the content as possible. Hence, two metrics are important, sentiment transfer ability and content retention. Following common practice (Hu et al., 2017; Shen et al., 2017; Lample et al., 2019), we measure the former with a separately trained style classifier based on DistilBERT (Sanh et al., 2019), and content retention in terms of self-BLEU (Papineni et al., 2002) between the input and the predicted output. Sometimes, comparing models is easiest via a single value. Therefore, others have argued to aggregate content retention and transfer accuracy metrics (Xu et al., 2018; Krishna et al., 2020). We follow the argumentation of Krishna et al. (2020) that this aggregation should be per sentence and compute a single $score = \frac{1}{M} \sum_{i=1}^{M} \text{ACC}(\hat{y}) \cdot \text{BLEU}(\hat{y}, x)$ where $x$ is the input sentence, $\hat{y}$ is the predicted sentence, and $M$ is the number of data points. For readability, we multiply all metrics by 100 before reporting.

As is standard practice in summarization, we evaluate performance on this task with ROUGE-L (Lin, 2004). Note, however, that ROUGE scores can be misleading if summaries are longer than the gold summary (Napoles et al., 2011). For this reason, we also report the average output length.

## 5.1 YELP-REVIEWS

Our hypothesis is that AEs with a single vector bottleneck are unable to reliably compress the text when it is too long. Here, we test if this holds true even for a large single-vector model with $d{=}512$. To this end, we create the dataset *Yelp-Reviews*, which consists of strongly positive and strongly negative English restaurant reviews on Yelp (see appendix A.3.1 for a detailed description). This dataset is very similar to Yelp-Sentences introduced by Shen et al. (2017). However, while Yelp-Sentences consists of single sentences of about 10 words on average, Yelp-Reviews consists of entire reviews of 52 words on average. To assess the reconstruction ability of BoV-AEs and fixed-sized AEs, we train 5 models that only differ in the target ratio. Models named **L0-r** denote L0Drop-based BoV-AE models with a target ratio $r$. The single vector fixed-size AE is obtained by averaging the representations at the last layer of the encoder. For style transfer, we train a $\text{Transformer}{+}{+}$ mapping using the loss described in Section 2. To obtain results at varying transfer levels, we train multiple times with varying $\lambda_{sty}$, resulting in multiple points for each model in Figure 3 and 5. Further details can be found in the appendix A.3.2.

**Results:** The results (full graph shown in Figure 7 in the appendix) indicate that even large single vector models ($d{=}512$) are unable to compress the text well; the NLL loss on the validation set of the fixed-size model is $\approx$3.9. **L0-0.05** is only slightly better than the fixed-size model, whereas **L0-0.1** already reaches a substantially lower reconstruction loss ($\approx$2.1). We evaluated the downstream sentiment transfer performance of $\text{Transformer}{+}{+}$ with **L0-0.1** [3] and the fixed-size model, respectively. Figure 3 shows a scatter plot of the results, where results that are further to the top-right corner are better. We see that at a comparable transfer level, the BoV is substantially better at retaining the input content. This supports hypothesis **H1** that variable-size BoV models are particularly beneficial in cases where the text length is too long to be encoded in a single-vector.

## 5.2 YELP-SENTENCES

In order to answer research question **H2**, we perform a large set of controlled experiments over our model's components. Due to the high computational demand, we turn to the popular Yelp-Sentences sentiment transfer dataset by Shen et al. (2017). Texts in this dataset are $\approx$ 10 words on average. As these sentences are much easier to reconstruct, we set the embedding size to $d{=}32$ so that the condition for hypothesis **H1** is still valid. Here, we again train BoV-AEs for target rates $r = 0.2, 0.4, 0.6, 0.8$ and then evaluate their reconstruction and style transfer ability in the same fashion as for Yelp-Reviews. Finally, we investigate the impact of the window size and differentiable Hausdorff loss. Experimental details are given in appendix A.4.2. For completeness, we provide an

---

[2]Pretraining a model of this size until convergence took more than a month on a single 24GB GPU.

[3]We restrict our analysis to L0-0.1 because this dataset have is computationally demanding.

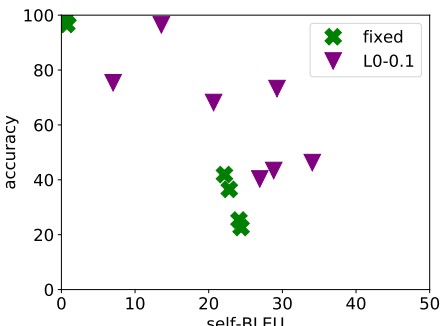

Figure 3: Style transfer on Yelp-Reviews.

Table 1: Results on Gigaword sentence summarization. Scores represent ROUGE-L with average output words in parentheses. T and T++ denote Transformer and Transformer++, respectively.

| Model | T | T++ |
|---|---|---|
| fixed | 13.1 (*18.3*) | 13.2 (*17.6*) |
| L0-0.2 | 19.8 (*23.2*) | 18.3 (*10.7*) |
| L0-0.4 | 8.0 (*18.7*) | 16.4 (*12.5*) |
| L0-0.6 | 6.6 (*83.5*) | 14.7 (*51.1*) |
| L0-0.8 | 9.3 (*5.1*) | 13.2 (*48.6*) |

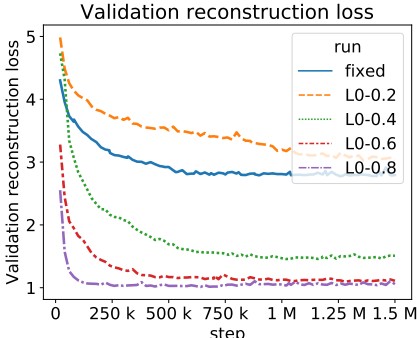

Figure 4: Reconstruction loss on the validation set for different AEs. **fixed**: The bag consists of a single vector obtained by averaging the embeddings at the last layer of the Transformer encoder. **L0-r**: BoV-AEs with L0Drop target ratio $r$.

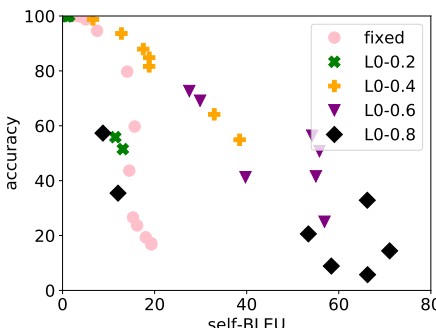

Figure 5: Style transfer performance on Yelp-Sentences of BoV models compared to a fixed-size AE for varying $\lambda_{sty}$. As both content retention and style transfer are important for style transfer, the further a graph is to the top right, the better the model.

analysis of the computational complexity of BoV-AE in Appendix B.2.2, and a qualitative analysis in Appendix B.2.3.

**Reconstruction ability:** Figure 4 shows the reconstruction loss on the validation set for the fixed-size model compared to BoV models. The fixed-size AE does not reach satisfactory reconstruction ability, converging at an NLL loss value of about 3. In contrast, BoV models are able to outperform the fixed-size model considerably. As expected, higher target ratios lead to better reconstruction, because the model can use more vectors to store the information. Models with a higher target ratio also reach their optimal loss value more quickly. While **L0-0.6** approaches the best reconstruction value ($\approx$1.0) eventually, the model needs more than 1 million training steps to reach it. In contrast, **L0-0.8** needs less than 100k steps to converge, which could indicate that **L0-0.8** learns to copy rather then compress the input, resulting in a bad latent space. **L0-0.4** yields to a higher loss, but is still drastically better than the fixed size model. **L0-0.2** is not enough to outperform the fixed-size model. Overall, these results show we have the right settings for evaluating **H1** and **H2**, as 10 words is too long to be encoded well into a single vector of $d$=32, whereas a BoV-AE with a high enough target ratio $r$ can fit it well.

**Style transfer ability:** Results are shown in Figure 5. Up to $r$=0.6, they correspond well to the reconstruction ability, in that BoV models with higher target ratios yield higher self-BLEU scores at comparable transfer abilities, outperforming the fixed-size model (**H1**). However, at $r$=0.8, the performance suddenly deteriorates at medium to high transfer levels. This supports the hypothesis that **L0-0.8** lacks smoothness in the embedding space due to insufficient regularization, which in turn complicates downstream training. This is the first piece of evidence that L0Drop is necessary for the success of our model (**H2**).

**Window size:** The window size $k$ determines which bag sizes around the input bag size we back-propagate from. Here, we investigates its influence on the model's performance. Since the $\lambda_{sty}$

hyperparameter is very sensitive to other model hyperparameters, we train with varying $\lambda_{sty}$ for each fixed window size and report the best style transfer score for each window size. Further experimental details and full results can be found in appendix A.4.3. Our results indicate that increasing the window size from zero (score 22.7) is beneficial up to some point ($k$=5, score 32.6), whereas increasing by too much ($k$=20, score 15.6) is detrimental to model performance even compared to a window size of zero. We hypothesize that backpropagating bags that are either very small or very large is detrimental because it forces the model to adjust its parameters to optimize unrealistic bags, taking away capacity for fitting realistic bags.

**Differentiable Hausdorff:** In Section 4.3, we argue that the min operation should be replaced by softmin in order to facilitate backpropagation. Here, we test if the differentiable version is really necessary, that is, we compare Eq. 6 to Eq. 7. Like before, we train the two variants with different $\lambda_{sty}$ and select the best style transfer score on the validation set. Further experimental details can be found in appendix A.4.3. The difference is substantial: Average Hausdorff reaches 14.6, whereas differentiable Hausdorff reaches 24.2. We hypothesize that this discrepancy is due to the difficult nature of the style transfer problem, which requires carefully balancing the two objectives, content retention (via Hausdorff) and style transfer (via the classifier). This is easier when the objective functions are smooth, which is the key advantage of differentiable Hausdorff.

## 5.3 SENTENCE SUMMARIZATION

In sentence summarization (Rush et al., 2015), the goal is to capture the essence of a sentence in fewer words. We evaluate on the Gigaword corpus (Graff et al., 2003) similar to Rush et al. (2015). This corpus consists of more than 8.5 million training samples, but we use a random subset of 500k to limit the computational cost. Inputs are on average 27 words long, which is medium length compared to the other two datasets in this study. We use moderately sized vectors of $d$=128 and again train different BoV-AEs with target ratios $r = 0.2, 0.4, 0.6, 0.8$. When applying the model to the sentence summarization downstream task, we train using the loss term $\mathcal{L}(\hat{\mathbf{z}}_y) = \mathcal{L}_{sim}(\mathbf{z}_x, \hat{\mathbf{z}}_y) + \lambda_{len}\mathcal{L}_{len}(\hat{\mathbf{z}}_y)$. This loss term is conceptually similar to the loss term used for style transfer, except that $\mathcal{L}_{len}$ denotes the prediction of a model trained to predict the length of the input text from the text's latent representation (the shorter the better). We train with varying values of $\lambda_{len} = 0.1, 0.2, 0.5, 1, 2, 5, 10$ and select the best model (ROUGE-L) on the development set. Intuitively, this model learns to retain as much from the input as possible while minimizing the output length. Note that this model of summarization could certainly be improved further, e.g. by accounting for relevancy and informativeness of the output (Peyrard, 2019). However, our goal is not to create the best task-specific model possible, so these considerations are out of scope for this paper.

The input texts in this task are relatively long. Due to the higher number of vectors in a BoV, it may be difficult to learn the mapping, especially for large target ratios $r$. We experiment with Transformer++ to observe to what extent this can facilitate learning.

**Results:** Despite the moderately large vector dimensionality, the single-vector bottleneck model achieves only considerably lower reconstruction performance than the BoV models (for details, please refer to Figure 8 in the appendix). Again, larger target rates $r$ lead to faster convergence, and all BoV models converge to approximately the same validation loss value (0.9). The only exception is **L0-0.2**, which converges to a higher loss value (1.25), but is still vastly stronger than the fixed size model (3.01).

However, as shown in Table 1, **L0-0.2** performs the best on the downstream task, outperforming the single-vector model by more than 5 ROUGE-L points while simultaneously requiring much fewer output words. BoV models with higher target ratios than $r$=0.2 perform worse. Moreover, the Transformer++ architecture tends to improve results, particularly with target rates $r > 0.2$. The ROUGE-L score itself does not improve for $r$=0.2, but note that this comes at the expense of more than doubling the output length. Also note that **L0-0.6** and **L0-0.8** only obtain relatively high scores because they produce long outputs that even exceed the length of the input. In fact, for $r = 0.6, 0.8$ no value of $\lambda_{len}$ produces outputs that are reasonably good ($> 10$ ROUGE-L) and short ($< 20$ BLEU) at the same time.

The above results confirm both our hypotheses: First (**H1**), it is beneficial to use a BoV model over a single-vector model to reduce the compression issues induced by the fixed-size bottleneck. Secondly (**H2**), when using a BoV model, it is imperative to regularize the number of vectors in the bag as

a way of smoothing the embedding space, making it easier to learn the mapping for unsupervised text generation tasks. Moreover, if the number of vectors in the bag is large, our Transformer++ architecture can substantially facilitate learning the mapping.

## 6 RELATED WORK

**Manipulations in latent space:** Besides Emb2Emb, latent space manipulations for textual style transfer are performed either via gradient descent (Wang et al., 2019; Liu et al., 2020) or by adding constant style vectors to the input (Shen et al., 2020; Montero et al., 2021). In computer vision, discovering latent space manipulations for image style transfer has recently become a topic of increased interest, in both supervised (Jahanian et al., 2020; Zhuang et al., 2021) and unsupervised ways (Härkönen et al., 2020; Voynov & Babenko, 2020). While these vision methods are similar to Emb2Emb conceptually, they differ from our work in important ways. First, they focus on the latent space of GANs (Goodfellow et al., 2014), which work well for image generation but are known to struggle with text (Caccia et al., 2020). Secondly, images typically have a fixed size, and consequently their latent representations consist of single vectors. Our work focuses on data of variable size, which may have important insights for modalities other than text, e.g. videos and speech.

**Unsupervised conditional text generation:** Modern unsupervised conditional text generation approaches are based on either **(a)** language models (LMs) or **(b)** autoencoders (AEs). **(a)** One type of LM approach explicitly conditions on attributes during pretraining (Keskar et al., 2019), which puts restrictions on the training data that can be used for training. Another type adapts pretrained LMs for conditional text generation by learning modifications in the embedding space (Dathathri et al., 2020). These approaches work well because LMs are pretrained with very large amounts of data and compute power, which results in exceptional generative ability (Radford et al., 2019; Brown et al., 2020) that even enables impressive zero-shot style transfer results (Reif et al., 2021). However, in contrast to AEs, LMs are not designed to have a latent space that facilitates learning in it. We therefore argue that AE approaches could perform even better than LMs if they were given equal resources. This motivates our research. **(b)** A very common approach to AE-based unsupervised conditional text generation is to learn a shared latent space for input and output corpora that is agnostic to the attribute of interest (e.g., sentiment transfer (Shen et al., 2017), style transfer (Lample et al., 2019), summarization (Liu et al., 2019), machine translation (Artetxe et al., 2018)). However, in these approaches, the decoder is explicitly conditioned on the desired attribute that must be available for all data points, complicating pretraining on unlabeled data. To overcome this, Mai et al. (2020) recently proposed Emb2Emb, which disentangles AE pretraining from learning to change the attributes via a simple mapping. Our paper makes an important contribution by improving the expressivity of Emb2Emb through variable-size representations.

## 7 CONCLUSION

Our paper addresses a fundamental research question from representation learning: How do we learn text representations in such a way that NLP tasks, specifically conditional text generation, can be learned in the latent space? Previous work (Emb2Emb) uses single-vector bottleneck autoencoders, which we argued to be fundamentally limited in how much text they can capture. We presented Bag-of-Vectors Autoencoders whose latent representation grows in size with the input as in attention-based models. We proposed L0Drop regularization, Transformer++, and differentiable Hausdorff to facilitate training in its embedding space. Controlled experiments revealed that BoV-AEs perform substantially better when the text is too long to be encoded into a single vector, even of size $d = 512$.

Our study is fundamental in nature; we do not focus on any particular task. Instead, we systematically demonstrate the benefit of Emb2Emb with variable-size representations rather than fixed-sized representations via controlled experiments. However, our model is in principle fit for the future. Mai et al. (2020) showed that the Emb2Emb framework benefits immensely from unlabeled data. As such, given enough compute and data for large-scale pretraining, Bag-of-Vectors Autoencoders could have the potential to become a *Foundation Model* (Bommasani et al., 2021) like BERT and GPT-3. Our study paves the way for the application of BoV-AEs for unsupervised tasks by demonstrating how to learn in their latent space.

## ETHICS STATEMENT

**Applications**   The focus of our study is not any particular application, but concerns fundamental questions in unsupervised conditional text generation in general. Unsupervised applications are useful in scenarios where few annotations exists, which is particularly common in understudied low-resource languages (e.g. unsupervised neural machine translation (Kuwanto et al., 2021)). Of course, oftentimes unsupervised solutions perform worse than supervised ones, requiring extra care during deployment to avoid harm from potential mistakes.

Despite the fundamental nature of our study, we test our model on two concrete problems, **a)** text style transfer and **b)** sentence summarization. **a)** Style transfer has applications that are beneficial to society, such as expressing "complicated" text in simpler terms (*text simplification*) or avoiding potentially offensive language (*detoxification*), both of which are particularly beneficial for traditionally underprivileged groups such as non-native English speakers. However, the same technology can also be used maliciously by simply inverting the style transfer direction. In this paper, we decided to study sentiment transfer of restaurant reviews as a style transfer task. The reasons are primarily practical; deriving both from the Yelp dataset, we can study the effectiveness of our model on two datasets (sentences and full reviews) that are very similar in content but considerably different in length. On one hand, this allows us to demonstrate the effectiveness of our model in a realistic, but computationally demanding setting. On the other hand, we can perform ablations in a less expensive setting. Apart from serving as a test bed for scientific research, sentiment transfer itself has no obvious real-world application. With enough imagination one can construe a scenario where a bad actor hacks into the database of a review platform like Yelp to e.g. manipulate the content of existing reviews. However, we rate this as highly unrealistic due to high opportunity cost, as it is much easier to generate fake reviews with large language models rather than hack into a system and alter existing reviews.

**b)** Summarization systems can be very valuable for society by enabling people to process information faster. But this depends on the system's output to be mostly factual, which neural summarization systems struggle with (Maynez et al., 2020). Unfaithful outputs may convey misinformation, which can potentially harm users.

**Deployment**   While we argue above that sentiment transfer has no useful real-world application, the model can still be deployed for demonstration purposes, or be trained and deployed for other tasks, e.g., sentence simplification. However, we urge not to deploy the models developed in this paper directly without adaptation for several reasons. i) The absolute performance is suboptimal (e.g., no large-scale pretraining) and hence makes many mistakes that a real-world application should avoid to prevent harm. ii) The model can occasionally produce toxic output. Of course, the extent to which this happens strongly depends on the training data. E.g., Yelp restaurant reviews can sometimes contain vulgar language. Any real-world application should hence consider pre- and post-filtering methods. iii) The model might be biased towards certain populations, the extent of which is not the subject of this study. For example, the sentiment transfer models would likely work better for fast food restaurants than restaurants of African cuisine, because the former is more common in the mostly US-centric data that the model is trained on. A real-world application needs to consider the requirements of the target audience.

Similarly, we argue that the sentence summarization model studied in this paper needs further improvements before deployment, some of which we mentioned in the main paper. Large-scale pretraining could also help to mitigate hallucinated facts (Maynez et al., 2020).

**Dataset**   The Yelp-Reviews dataset is a direct derivative of the Yelp Open Dataset[4]. Their license agreement states that any derivative remains the property of Yelp, hence we can not directly release the dataset. However, academic researchers can easily obtain their own license for non-commercial use and recreate the dataset used in this study via the script we provide in the supplementary material. No further data collection was conducted.

---

[4]`www.yelp.com/dataset`

## REPRODUCIBILITY STATEMENT

We took several precautions to ensure that our work is reproducible.

**Datasets**   Our study is based on two existing datasets, Gigaword sentence summarization, and Yelp-Sentences style transfer. For these two datasets, we provide scripts that preprocess them as in our study. For Yelp-Reviews dataset, we provide a detailed description in appendix A.3.1. Moreover, we provide a script that allows to construct the dataset as a derivative from Yelp data. In order to get access to Yelp data, practitioners have to obtain a license from Yelp that is free of charge. The data may only be used for non-commercial or academic purposes, but this suffices to reproduce our study. The Gigaword corpus is commonly used, and can be downloaded from the Linguistic Dataset Consortium at `https://catalog.ldc.upenn.edu/LDC2012T21`. For downloading, a membership is mandatory, or otherwise fees apply. However, this commonplace in NLP research institutes.

**Code**   We provide anonymized code to reproduce all our experiment in the supplementary materials. Upon acceptance, the code will be made available to the public.

**Experiments**   We provide details on each experiment's setup in the appendix. However, it's impractical to report all details that may impact the outcome. Therefore, for each experiment we additionally provide a csv file in the supplementary material. The file contains information on all training parameters, model hyperparameters and results. In combination with the code, this allows to reconstruct almost the exact experimental setup used in our study apart from parameters that are beyond our control, such as the computation environment.

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

# Appendix

## A    EXPERIMENTAL DETAILS

Here, we describe the experimental setup used in our experiments. We try to be exhaustive, but the exact training configurations and code will also be given as downloadable source code for reference.

Table 2: Basic statistics for each dataset used in this study. Average number of words refers to input texts and output texts, respectively.

| Dataset | avg. #words | #inputs | #outputs |
|---|---|---|---|
| Yelp-Sentences | 9.7 / 8.5 | 177k | 267k |
| Gigaword | 27.2 / 8.2 | 500k | 500k |
| Yelp-Reviews | 56.1 / 48.7 | $500k$ | $500k$ |

### A.1    AUTOENCODER PRETRAINING

All autoencoders consist of standard Transformer encoders and decoders (Vaswani et al., 2017), with 3 encoder and decoder layers, respectively. The Transformers have 2 heads and the dimensionality is set to the same as the latent vectors (Yelp-Reviews: 512, Yelp-Sentences: 32, Gigaword: 128). In case of the fixed sized model, the representations at the last layer are averaged. Otherwise we perform L0Drop as described in Section 3. We set $\lambda_{L_0} = 10$ for all BoV models and only vary the target ratio. All models are trained with a dropout (Srivastava et al., 2014) probability of $0.1$ and a denoising objective, i.e, tokens have a chance of 10% to be dropped from the sentence. We train the model with the Adam optimizer (Kingma & Ba, 2015) with an initial learning rate of $lr = 0.00005$ (Yelp-Reviews and Gigaword) or $lr = 0.0001$ (Yelp-Sentences) and a batch size of 64. We experimented with other learning rates $(0.00005, 0.0005)$ for the fixed-size model on Yelp-Reviews, but the results did not improve. Models are trained for 2 million steps on Gigaword and Yelp-Reviews and for 1.5 million steps on Yelp-Sentences. We check the validation set performance every 20,000 steps and select the best model according to validation reconstruction performance.

### A.2    DOWNSTREAM TASK TRAINING

After the autoencoder pretraining, we train downstream by freezing the parameters of the encoder and decoder. The dimensionality of the one-layer mapping $\Phi$ (a Transformer decoder with 4 heads) is set to the same as the latent representation (Yelp-Reviews: 512, Yelp-Sentences: 32, Gigaword: 128). We set the maximum number of output vectors to $N = 250$ on Yelp-Reviews and Gigaword, and $N = 30$ on Yelp-Sentences. The batch size is 64 for Yelp-Sentences and Gigaword and 16 on Yelp-Reviews. We train for 10 epochs on Yelp-Sentences and Gigaword, and for 3 epochs on Yelp-Reviews. The validation performance is evaluated after each epoch.

**Losses:** In all tasks we have two loss components. For $\mathcal{L}_{sim}$, we use differentiable Hausdorff unless specified otherwise (in the ablation). $\mathcal{L}_{sty}$ and $\mathcal{L}_{len}$ depend on classifiers / regressors, which we train separately after the autoencoder pretraining as a one-layer Transformer encoder. The embeddings are then averaged and plugged into a one-layer MLP whose hidden size is half of the input size and uses the tanh activation function. These classifiers are trained via Adam ($lr = 0.0001$) for 10 epochs and we evaluate the validation set performance after each. The total loss depends on a window size as described in Equation 4. For performance reasons (multiple computations of the loss), we set $k = 0$ unless specified differently.

### A.3    YELP-REVIEWS

#### A.3.1    DATASET

The dataset was obtained from `https://www.yelp.com/dataset` in May 2021. Our goal is to obtain texts long enough such they cannot be reconstructed by a reasonably sized autoencoder with a single-vector bottleneck. We find that to be the case when limiting ourselves to reviews of

maximum 100 words. We apply this limit due to the computational complexity of Transformers on long texts. Otherwise, we stick with similar filtering criteria as Shen et al. (2017): We only consider restaurant businesses. We consider reviews with 1 or 2 stars as negative, and reviews with 5 stars as positive. We don't consider reviews with 3 or 4 stars to avoid including neutral reviews. We subsample 400,000 positive and negative reviews for training, respectively, and use 50,000 for validation and test set each.

In order to demonstrate the usefulness of our model on long texts, we turn to the original Yelp dataset[5]. Our goal is to obtain texts long enough such they cannot be reconstructed by a reasonably sized autoencoder with a single-vector bottleneck. We find that to be the case when limiting ourselves to reviews of maximum 100 words[6]. Otherwise, we stick with similar filtering criteria as Shen et al. (2017): We only consider restaurant businesses. We consider reviews with 1 or 2 stars as negative, and reviews with 5 stars as positive. We don't consider reviews with 3 or 4 stars to avoid including neutral reviews. We subsample 400,000 positive and negative reviews for training, respectively, and use 50,000 for validation and test set each.

### A.3.2 DOWNSTREAM TRAINING

For both the fixed-size model and the BoV model (**L0-0.1**), we choose the best learning rate among $lr = 0.0001$ and $lr = 0.0005$ on the validation set and report test set results. We train with $\mathcal{L}_{sty} \in \{0.1, 0.2, 0.5, 1, 2, 5, 10\}$, resulting in the scatter plot in Figure 3.

### A.4 YELP-SENTENCES

### A.4.1 DATASET

Yelp-Sentences consists of the sentiment transfer dataset created by Shen et al. (2017), who made their data available at `https://github.com/shentianxiao/language-style-transfer/tree/master/data/yelp`. We use their data as is without further preprocessing. Table 2 presents some basic statistics about this dataset.

### A.4.2 DOWNSTREAM TRAINING

We train BoV models with $\lambda_{sty} \in \{1, 2, 5, 10, 20, 50, 100\}$. To make sure that our results are not due to insufficient tuning, for the fixed-sized model, we use the following larger range: $\{0.01, 0.02, 0.05, 0.1, 0.2, 0.5, 1, 2, 5, 10, 20, 50, 100\}$. All configurations are trained with $lr = 0.0005$. These results produce the scatter plot in Figure 5.

### A.4.3 ABLATIONS

For the ablations on differentiable Hausdorff distance and the window size, we use the **L0-0.6** model. For each option, we train with $\mathcal{L}_{sty} \in \{0.1, 0.2, 0.5, 1, 2, 5, 10, 20, 40, 60, 80, 100\}$ and report the best value in terms of style transfer score on the validation set.

### A.5 SENTENCE SUMMARIZATION

### A.5.1 DATASET

The dataset is based on the Gigaword corpus (Graff et al., 2003). We largely follow the preprocessing in Rush et al. (2015), which we obtained from the paper's GitHub repository at `https://github.com/facebookarchive/NAMAS`. Different from them, we convert all inputs and outputs to lower case and use a smaller split (1 million examples). We provide the scripts for constructing the dataset from a copy of the Gigaword corpus (which can be obtained from the Linguistic Dataset Consortium) together with the rest of our code.

---

[5]The dataset was obtained from `https://www.yelp.com/dataset` in May 2021.

[6]We apply this limit due to the computational complexity of Transformers on long texts.

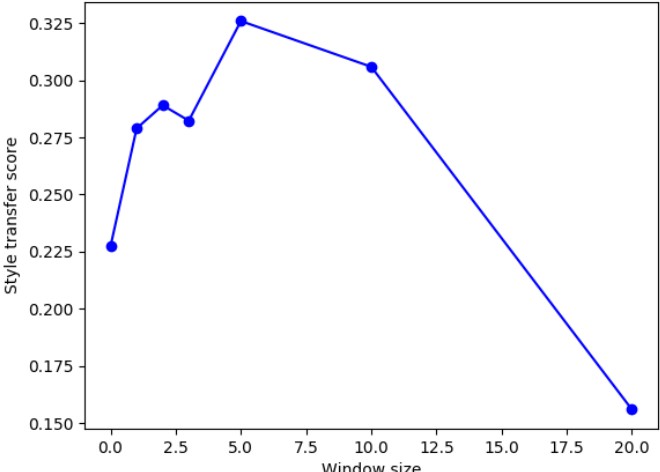

Figure 6: Style transfer score depending on the window size.

### A.5.2 DOWNSTREAM TRAINING

We train all models with $lr = 0.00005$. For each target ratio $r$ and each of Transformer and Transformer++, we select the best $\lambda_{len} \in \{0.1, 0.2, 0.5, 1, 2, 5, 10\}$ in terms of ROUGE-L on the validation set and report test set results in Table 1.

## B ADDITIONAL RESULTS

### B.1 YELP-REVIEWS

In Figure 7, we plot the reconstruction ability of the fixed-size model compared to the BoV-AEs on the validation set.

Again, despite a large dimensionality ($d = 512$), the single-vector model achieves substantially lower reconstruction ability than BoV-AE. With respect to the target sparsity rate, we find that $r = 0.1$ is enough to reach dramatically better results than the fixed-size model, whereas $r = 0.05$ only reaches slightly better results after two million training steps. However, the plot shows clearly that **L0-0.05** has not converged, suggesting that **L0-0.05** could reach much better performance if trained for even longer.

### B.2 YELP-SENTENCES

### B.2.1 WINDOW SIZE

In Figure 6, we plot the style transfer score as a function of the window size. As already described in Section 5.2, the performance improves almost monotonically up to a certain point, after which the performance decreases again. This demonstrates the importance of choosing the right window size.

### B.2.2 COMPUTATION TIME

Our experiments have shown that bag-of-vector representations are more powerful than single-vector representations. However, the increased capacity of BoV-AE comes at the expense of higher computation time. The size of the latent representation impacts the computation time in two places: During cross-attention in the decoder and when computing the mapping. Asymptotically, the decoder's cross-attention mechanism computes $\mathcal{O}(n \cdot |s|)$ dot-products, where $n$ is the number of vectors in the latent representation and $|s|$ is the length of the text sequence $s$. When computing the mapping,

Table 3: Asymptotic computation time in the Emb2Emb framework as a function of the latent representation size $n$ and the length of the input text $|s|$, depending on the type of autoencoder.

| AE type | Cross-Attention Decoding | Mapping |
|---|---|---|
| in general | $\mathcal{O}(n \cdot |s|)$ | $\mathcal{O}(n^2)$ |
| fixed | $\mathcal{O}(|s|)$ | $\mathcal{O}(1)$ |
| BoV-AE | $\mathcal{O}(|s|^2)$ | $\mathcal{O}(|s|^2)$ |

Table 4: The number of seconds it takes to process 5% of the validation set (1264 samples) with a batch size of 1. Lower is better.

| Model | Encoding | Mapping | Decoding |
|---|---|---|---|
| fixed | 4.8 | 2.4 | 51.7 |
| L0-0.4 | 7.2 | 12.6 | 50.1 |
| L0-0.8 | 7.3 | 20.6 | 50.3 |

both at training and inference time, we produce a fixed number $N$ of vectors autoregressively, but in most applications, $N$ can reasonably be bound by a linear function of $n$ (e.g., $2n$ in style transfer or $0.5n$ in summarization). The mapping is essentially a Transformer decoder, so both the cross attention and self attention parts compute $\mathcal{O}(n^2)$ dot-products. Given that $n = 1$ for single-vector AEs and $n = \mathcal{O}(|s|)$ for BoV-AEs with L0Drop, we obtain the asymptotic complexities as shown in Table 3.

To assess the empirical impact, we measure the wallclock time of Emb2Emb's "Inference" stage (cf. Figure 2). We take separate measurements for encoding, mapping, and decoding, respectively. Since decoding speed depends on the quality of generation (e.g., when the end-of-sequence symbol is generated late due to repetitions), we do the following to enable fairer comparisons. We enforce the same fixed number of decoding steps (10) in all models. The mapping is set to produce as many output vectors as input vectors. We use a batch size of 1, but note that the results would largely extend to larger batch sizes when binned batching is used. The results are shown in Table 4.

Both the encoding and the mapping stages of Emb2Emb are more expensive in BoV models than in the fixed-size model. The difference in the encoding stage can be explained by the overhead through the L0Drop layer, which includes identifying near-zero gates and discarding their respective vectors. The difference in the mapping grows with higher L0Drop target ratios. This is expected since the number of autoregressive steps decreases with the target ratio. Finally, we do not observe any meaningful speed differences between the models at decoding time. This is somewhat surprising, but could be explained by two factors. First, the *self-attention* part of the decoder already has a complexity of $\mathcal{O}(|s|^2)$, which probably dominates the total computation time. Secondly, the computation of the dot-product is easy to parallelize. In summary, we find that BoV models are slower overall, especially in the mapping. However, since our L0Drop implementation prunes near-zero vectors, lower target rates mitigated the additional computation overhead. This is especially evident when comparing training speeds. While **L0-0.8** processes 15 sentences per second, **L0-0.4** processes can process 21 (fixed-size: 42).

### B.2.3 QUALITATIVE ANALYSIS

Standard autoencoders suffer from poor performance with Emb2Emb if the text is too long to be encoded into a single vector. BoV-AEs were designed to alleviate this issue. Here, we conduct a qualitative analysis of 10 randomly selected model outputs on Yelp-Sentences. For comparability, we select models with similar levels of style transfer accuracy, namely the fixed size model with a performance of 59% accuracy and 17 points self-BLEU to **L0-0.4** with a performance of 55% accuracy and 38 points self-BLEU. We randomly sample 10 examples and show them in Table 5. By design of the Yelp-Sentences dataset (Shen et al., 2017), the input sentences are sentences drawn from negative reviews, whose sentiment are supposed to be changed to positive. Note that due to how the dataset was constructed, some of the input sentences are already positive (#7) or just neutral (#2).

We observe several trends:

| # | Input sentence | Output of fixed-size model | Output of L0-0.4 |
|---|---|---|---|
| 1 | generally speaking it was nothing worth coming back to . | but there here here and it will enjoy it . | generally remain it was it worth it and always happy ! |
| 2 | then why did n't they put some in ? | then she , you ta are the in the ? | then ' why n ' t they put some delicious ! |
| 3 | horrible experience ! | horrible ! | horrible experience ! |
| 4 | it was a shame because we were really looking forward to dining there . | it was a a fun , there and we have been to . | it really nice shame because we were really looking forward forward and fantastic ! |
| 5 | suffice to stay , this is not a great place to stay . | suffice to to not stay to this place is a stay . | suffice is not stay , this is a great place and always great ! |
| 6 | the chicken was weird . | the chicken was weird . | the chicken was weird . |
| 7 | my mom ordered the margarita panini which was pretty good . | my my margarita was ordered which was very good . | my mom ordered the margarita panini which was pretty good . |
| 8 | i 'm not willing to take the chance . | i will definitely recommend your time or you . | i ' m not willing to take the great . |
| 9 | i would say for the price point that it was uninspired . | i had this place at the food , it 's super . | i would say for the price point that it was delicious . |
| 10 | the only pool complaint i have was from the last day of our stay . | the waitress was the the the the time here a last time | the only pool complaint i have was from the day was wonderful ! |

Table 5: 10 randomly sampled examples from Yelp-Sentences.

1. The fixed-sized model has a difficult time retaining the aspect discussed in the input sentence (#10: staff instead of location, #9: food instead of price), whereas the BoV-AE stays on topic. This is likely a consequence of the fixed-sized model's inability to encode the input well into a single vector.

2. The outputs of the fixed-sized models are often completely unusable (#1, #2) or nonsensical (#5, #9, #10), whereas the outputs of the BoV-AE are at least intelligible.

3. In absolute terms, the outputs of neither model are reliably grammatical or able to flip the sentiment. This is understandable since no large pretrained language model is used. We would like to stress that therefore our models should not be used in production as they are (cf. Ethics Statement). Large-scale pretraining is needed to produce coherent outputs (Brown et al., 2020), which then produces impressive outputs on style transfer (Reif et al., 2021). As we argue in Section 1 and 7, our paper contributes to the foundation for large scale pretraining of autoencoder models to be used in Emb2Emb.

## B.3 SENTENCE SUMMARIZATION

Figure 8 shows the development of the reconstruction loss on the validation set over the course of 2 million training steps. Despite a moderately sized dimensionality ($d = 128$), the single-vector bottleneck model achieves only considerably lower reconstruction performance than the BoV models. Moreover, we can see again that larger target rates $r$ lead to faster convergence, and all BoV models converge to approximately the same validation loss value.

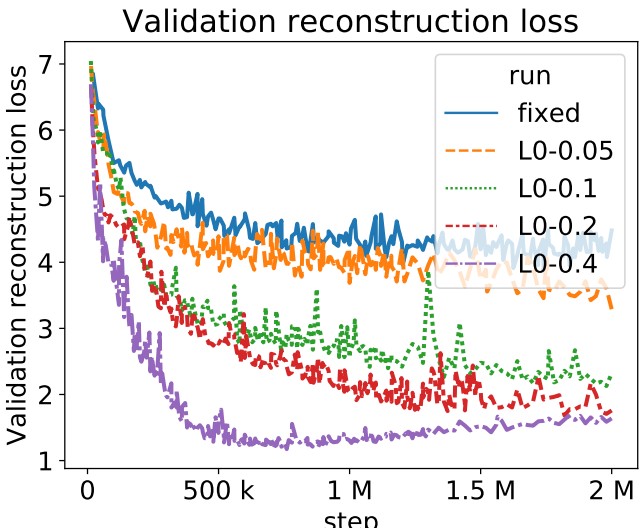

Figure 7: Reconstruction loss on the validation set of Yelp-Reviews for different autoencoders. **fixed**: The bag consists of a single vector obtained by averaging the embeddings at the last layer of the Transformer encoder. **L0-r**: BoV-AE with L0Drop target ratio $r$.

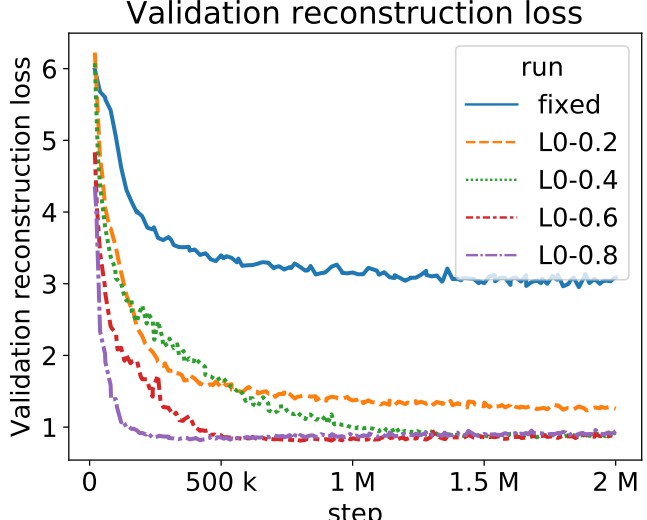

Figure 8: Reconstruction loss on the validation set of Gigaword for different autoencoders. **fixed**: The bag consists of a single vector obtained by averaging the embeddings at the last layer of the Transformer encoder. **L0-r**: BoV-AE with L0Drop target ratio $r$.

