# OpenReview forum: "Bag-of-Vectors Autoencoders for Unsupervised Conditional Text Generation"
_ICLR.cc/2022/Conference — ICLR 2022 Submitted_

### Official Review · Reviewer_kHZy · 2021-10-31

**Correctness:** 3
**Technical Novelty And Significance:** 3
**Empirical Novelty And Significance:** 3
**Recommendation:** 6
**Confidence:** 3

**Main Review:**

Strengths:

- This is a technically sound submission which appears to be novel.
- The idea is relatively simple and intuitively appealing. It makes sense that the representational capacity of a fixed-sized embedding may be limited, and therefore utilizing variable-sized embeddings is a natural extension.
    - Figures 1 and 2 are clear in describing the proposed method at a high level.
- The empirical results on style transfer and summarization appear to be strong (although as mentioned below, Figures 3 and 5 are unclear).

Weaknesses:

- The writing is unclear in a number of places.
    - Figure 2
        - Caption mistakenly says \hat{\mathbf{z}}_x instead of \hat{\mathbf{z}}_y in part called "Task Training".
    - Equation (2)
        - X should be replaced by \mathbb{X}
    - Section 4.1 paragraph 3
        - It is slightly confusing to say that \mathbf{z}_t' is the output of the Transformer. Maybe it would be clearer to actually use the notation \mathbf{z}_t' in Equation (2) and then define it immediately thereafter?
    - Section 4.2 paragraph 2
        - It is not especially clear what is meant by "we minimize the loss locally at every step". Presumably, as described in Section 2, this is task dependent and perhaps something like the loss described in Section 2 paragraph 2, but it would be useful to the reader if this were described somewhat more explicitly.
    - Section 5.1 paragraph 2
        - Should the Figure being referred to be 7 and not 8?
    - Section 5.3 paragraph 1
        - It would be useful to add a short description (or citation) to explain what a length regressor is.
    - Figures 3 and 5
        - It is not clear what the different points for either fixed or each value of L0 represent?
        - Based on the score shown in Section 5 paragraph 3, why isn't there only one point for each model?
    - Some of the important equations would be better presented as blocks rather than inline, e.g. the loss in Section 2 paragraph 2, the score in Section 5 paragraph 3, etc.
- L0Drop
    - Suppose one of the gates g_i = 0, which then means \mathbf{z}_i = \mathbf{0}. In Figure 2 top, is the Transformer decoder still able to attend to this \mathbf{z}_i? And in Equations (2) and (3), is this \mathbf{z}_i a component of the computations (in particular, will it still have a non-zero contribution to the softmax both in the Transformer in Equation (2) and in the \alpha_i term in Equation (3))?
        - It could be argued that if so, the encoder is not outputting a bag of vectors whose size is smaller than the number of tokens, because a 0-vector is different from there not being a vector at all.
        - Ideally, this \mathbf{z}_i would be completely masked out during the decoding stage rather than being set to 0.
- Experiments
    - It would add a lot of value to the paper to see examples of the model outputs on sentiment transfer and sentence summarization.
    - A discussion on the computational efficiency compared to the baseline would also be very useful.

**Summary Of The Paper:**

This paper presents an autoencoding model which is trained for conditional text generation tasks in an unsupervised manner. Rather than using a fixed-sized bottleneck layer in the autoencoder, the authors present a method for learning a variable-sized set of representations (referred to as a bag of vectors). This bag of vectors can then be transformed (in embedding space) in order to generate a modified sequence of text based on the original input sequence.

**Summary Of The Review:**

Overall this is a good paper which, with some adjustments to the writing, would be of good value to the community.

---

> ### Author Response · Authors · 2021-11-21
> **Reply to review**
>
> Thank you for your thoughtful review. Your tips on improving the writing were very helpful, and we applied them in the paper draft. Here are some specific clarifications:
> * We added an example of the loss to the first paragraph of Section 4. Does this make the section clearer for you?
> * In text style transfer, two metrics are important, content retention (measured as self-BLEU) and style transfer accuracy (measured in terms of an external classifier). Figures 3 and 5 show multiple points per model (e.g. fixed-size and L0-0.2) because we trained multiple downstream models with varying $\lambda_{sty}$. We had already noted that in the first paragraph of 5.1, but we added a half sentence there to make the connection to the graphs clearer. We also added this information to the caption of Figure 5.
>
> Regarding your questions on L0Drop, we explicitly state (already in the initial draft) in 5.1 that vectors whose gates are zero or close to zero are pruned from the bag entirely. Hence, the decoder is unable to attend to them, and they are not considered in Equation 2 and 3.
>
> As you suggested, we added a speed comparison in Appendix B.2.2, and referenced it from Section 5.2 in the main paper. As expected, the single-vector baseline is computationally much more efficient than the BoV models, but applying a lower L0Drop target rate improves the efficiency of BoV models.
>
> We also added a qualitative analysis of outputs on Yelp-Sentences in Section 5.2.3. They confirm our hypothesis that the fixed-sized model is often just not able to compress the input well, as the outputs often deviate in terms of topic or wording from the input.

---

> > ### Comment · Reviewer_kHZy · 2021-11-29
> > **Reply to authors**
> >
> > Thanks for the responses; the writing is now much clearer. Given that the outputs in Table 5 aren't of particularly high quality, I still think that a weak accept is the appropriate score for this paper.

---

### Official Review · Reviewer_hEPH · 2021-11-02

**Correctness:** 4
**Technical Novelty And Significance:** 3
**Empirical Novelty And Significance:** 2
**Recommendation:** 5
**Confidence:** 3

**Main Review:**

STRENGTHS

- The paper considers the general problem of learning an encoder that encodes the input to a variable-length sequence. This can be potentially useful in settings beyond unsupervised conditional text generation.

- The proposed approach (e.g., using Hausdorff for alignment) is sensible and effective.

- BoV-AEs clearly outperform Emb2Emb on a variety of tasks and metrics.


WEAKNESSES

- The experiments do not involve any other baseline. The authors say upfront that the goal of the experiments is to confirm that the multi-vector extension improves over Emb2Emb and that their proposed techniques are necessary, but a total lack of comparisons with existing works in this area makes it difficult for the reader to contextualize the contribution of the work with respect to methods other than the authors' own.

- The writing is a bit unclear, though it can be followed with some efforts. For instance, the authors write "X = {z_1 ... z_n} := enc(x)" in the first paragraph of Section 3 without defining "n". It took me a while to guess that this is the input bag size that the user provides.

- I'm not an expert on this topic, but it seems unlikely that this is the only work that considers multi-vector autoencoding. See [1] for example. But there is none discussed.

[1] SEQ^3: Differentiable Sequence-to-Sequence-to-Sequence Autoencoder for Unsupervised Abstractive Sentence Compression (Baziotis et al., 2019)

**Summary Of The Paper:**

The paper extends the unsupervised conditional text generation framework of Emb2Emb (Mai et al., 2020) from a single encoding to variable-length encodings. The new model is called a bag-of-vectors autoencoders (BoV-AE). To train BoV-AEs, the paper develops a regularization technique called L0Drop, an extension of OffsetNet in Emb2Emb for learning the mapping Phi, and an alignment loss based on the Hausdorff distance. Experiments show that BoV-AEs outperforms Emb2Emb on sentiment transfer and summarization in the reconstruction loss, Rouge/Bleu, and accuracy.

**Summary Of The Review:**

The paper presents a reasonable multi-vector extension of Emb2Emb (Mai et al., 2020), but is a bit lacking in comparison with other existing works.

---

> ### Author Response · Authors · 2021-11-21
> **Reply to review**
>
> Thank you for your review. You raised the following concerns:
> * __“The experiments do not involve any other baseline”__:
> Our main goal is to extend the Emb2Emb framework to multi-vector embeddings, not to maximise performance on any particular task.  In this sense, the hypotheses we formulate and the experiments we use to test them are appropriate, and we don’t see any other baselines we could use.  To address the broader question of how to train autoencoders with multi-vector latent representations that are useful for learning in the embedding space, the lack of previous work on this topic makes it hard to imagine what the appropriate alternative models might be, as discussed further below.
> * __“The writing is a bit unclear”__: We applied your and other reviewers’ remarks on how to improve the clarity of writing. Please let us know if you have more actionable feedback regarding the clarity.
> * __“it seems unlikely that this is the only work that considers multi-vector autoencoding”__
> Of course a Transformer encoder-decoder architecture could be used as a multi-vector autoencoder. For example, BART [1] is pretrained with a multi-vector denoising autoencoder objective and subsequently finetuned. But our experiments with L0Drop show that denoising alone is not enough to make Emb2Emb training feasible - the number of vectors also has to be reduced. We show that L0Drop is effective at that, thus contributing to this broader issue. But note that we do not claim that L0Drop is the only way or best way to achieve this goal. While definitely a worthwhile question, it is out of scope for this paper, which focuses on how to learn sequence-to-sequence tasks in the embedding space of a multi-vector autoencoder (Emb2Emb). All our technical contributions are designed to facilitate learning in this embedding space. We discuss related work on latent space manipulations, which we consider the core of our paper, in Section 6. Our paper’s focus is NOT on other typical metrics of autoencoders such as reconstruction ability, generative abilities, or downstream classification tasks.
> You mention SEQ^3, which is different to our model because it is specifically designed for unsupervised sentence summarization, trained in an end-to-end fashion (therefore not applicable in the Emb2Emb setting), and has a discrete latent space that simultaneously functions as the summary during evaluation.

---

### Official Review · Reviewer_jHbb · 2021-11-02

**Correctness:** 3
**Technical Novelty And Significance:** 3
**Empirical Novelty And Significance:** 2
**Recommendation:** 5
**Confidence:** 4

**Main Review:**

Here are some suggestions and questions:
- I think the authors should describe more about the autoencoder. For example, is the input of the encoder is a bag of words or sequential tokens (a bag of words + positional encoding)? If it's the former case, why ignore orders? The order should provide lots of information about semantics. If it's the latter case, it is weird to view the embedded as a bag of vectors since the order indeed matters a lot.
- If the authors treat the latent representations as a bag of vectors (order doesn't matter), why does the transform function Phi uses autoregressive decoding to produce z_y? What is the order for autoregressive decoding? Is it the same order as the order of the original tokens? Also, if the order doesn't matter a lot, why not use Trasnforemer to generate all z_y at the same time?
- It would be great if the author can provide some other baselines, such as Mai et al. (2020), although they are using LSTM.
- Any ablation studies for the pointer-generator technique?
- From my perspective, if only replacing the autoencoder with a stronger one, the novelty is not enough. The important part should be handling the transform between multiple vectors instead of a single vector. However, there is a lot of unclear parts for handling the transform between multiple vectors when I read the paper.

**Summary Of The Paper:**

This paper extends previous work, Emb2Emb, by replacing the LSTM autoencoder with a Transformer autoencoder. The authors propose some techniques to handle the difficulty of this replacement. They also conduct some experiments to support the claim.

**Summary Of The Review:**

I think the novelty is not enough and some details should be described more clearly.

---

> ### Author Response · Authors · 2021-11-21
> **Reply to review**
>
> Thank you for your review. The reviewer seems to have misunderstood the contribution of our paper.  The important point is indeed handling multiple vectors instead of a single vector, hence the title “Bag-of-Vector Autoencoders ...”.  We will try to clarify the misunderstandings in our replies to specific questions below:
>
> * __“From my perspective, if only replacing the autoencoder with a stronger one, the novelty is not enough. The important part should be handling the transform between multiple vectors instead of a single vector.”__:
> The important part of our paper is explicitly how to handle the transform (Emb2Emb) between multiple vectors instead of a single vector. We state this in the title and the introduction, illustrated by Figure 1. This is a novel research question that requires novel technical contributions. Indeed, all of our major technical contributions (L0Drop, differentiable Hausdorff loss, and $\Phi$ being a modified pointer-generator network) specifically target the question of how to facilitate learning the mapping from one bag-of-vectors to another bag-of-vectors, which you say should be the important part. This question is so central to the paper that it receives its own explicitly stated hypothesis (H2) in the beginning of Section 5. If you still think that the paper lacks novelty or emphasis, please explain why the aspects mentioned above do not apply.
> * __“Any ablation studies for the pointer-generator technique?”__: You probably missed that the paper DOES contain an ablation study for the pointer-generator technique on summarization in Table 1. Those results are discussed in length in Section 5.3.
> * __“If [the input is ... a bag of words + position embeddings], it is weird to view the embedded as a bag of vectors since the order indeed matters a lot.”__: As is conventional with Transformers applied to text, the order of tokens is encoded at the input layer through the addition of positional embeddings. However, that does not mean that the latent representation itself is order-dependent, too. In fact, as we already explain in Section 3, position embeddings are precisely necessary because the order of vectors in the bag does not matter when processed via attention, such as  the cross-attention mechanism of the Transformer decoder. Moreover, due to the use of L0Drop there is no one-to-one relation between input tokens and vectors in the bag. The model is forced to learn to compress entire phrases into higher-level representations. In order to be able to reconstruct the input tokens in correct order, those higher-level representations will still contain order information. We don’t see how this contradicts the fact that the order of the vectors in the bag does not matter.
> * __“why does the transform function Phi uses autoregressive decoding to produce z_y?”__: Indeed, we use an autoregressive prediction model for $\Phi$ for simplicity. As you suggest, previous to submission we did consider producing all vectors at the same time using a Transformer encoder model. However, this has a major caveat: The size of the output bag would have to be the same as the input bag. This may be a good model for tasks where inputs and outputs are approximately the same length. But since we wanted to keep our model as general as possible (e.g. for application to summarization), we decided against this.
> Regarding the order in autoregressive decoding: Note that our loss functions are defined on bags-of-vectors, not on sequences of vectors: We DO NOT provide gold labels at every time step of the autoregressive predictions. Rather, we propose a differentiable variant of the modified Hausdorff distance loss, which is specifically designed for sets, NOT sequences. That is, given a bag-of-vectors predicted by $\Phi$, the loss is indifferent with respect to the order in which the vectors were generated.
> * __“However, there is a lot of unclear parts for handling the transform between multiple vectors when I read the paper”__:
> What are your specific questions? We already applied a couple of remarks from other reviewers on how to improve the writing. Without explicit actionable feedback from your side, we don’t know what to make of this comment.

---

> > ### Comment · Reviewer_jHbb · 2021-11-21
> > **Reply to response**
> >
> > Thanks for the response. Previously I got a bit confused because the description about "autoregressive decoding to produce z_y" and may have some misunderstanding. Now, I got the point of this paper, to handle Emb2emb for multiple vectors, it replaces $\Phi$ from MLP to Transformer (with autoregressive decoding) and proposes L0drop and Hausdorff loss to compute the distance between sets of vectors. I'll change my score accordingly.
> >
> > Here are some questions and suggestions I have now:
> > - To show the importance of using multiple vectors, I thought the original Emb2emb (Mai et al. (2020)) should be considered as a baseline for fixed-size AE as well. It uses LSTM so it should be much weaker than the proposed method. However, I found the figure 5 in Mai et al. (2020) seems to have better scores on yelp dataset than the proposed one. Is there any difference for the experimental setting?
> > - Why the fixed AE uses the average of representations of encoder, not the decoder with only one step autoregressive decoding? The number of parameters may affect the performance as well.
> > - Regarding the autoregressive decoding, I have the same though with Reviewer 74E4, the mapping seems not to be a real mapping between bag of vectors since the latter vectors depend on former vectors. How about training a decoder with k learnable input vectors? In that way, you can decode the k bag-of-vectors in parallel so it should be real bag-of-vectors.

---

> > > ### Author Response · Authors · 2021-11-21
> > > **Reply to additional questions**
> > >
> > > Thank you for your quick reply. To answer your questions:
> > > * __“I found the figure 5 in Mai et al. (2020) seems to have better scores on yelp dataset than the proposed one. Is there any difference for the experimental setting?”__ The hypothesis we are testing (H1 introduced in the beginning of Section 5 in the paper) is the following: If the text is too long to be encoded into a single vector of some dimensionality, BoV-AE provides a benefit over single-vector autoencoders. As shown in Table 2 of appendix A, Yelp-Sentences consists of very short texts of approximately 10 words. Such short sentences can almost perfectly be reconstructed by a fixed AE model with 512 dimensions, so there would be no difference between these models. However, in the experiments on Yelp-Sentences in our paper, fixed AE and BoV-AE used 32 dimensions per vector, which is not enough for a fixed AE to encode the text (see Figure 4). Setting it to 32 is thus necessary to fulfill the condition of our hypothesis (printed in bold font above). This difference also explains why the performance is worse than in Mai et al. (2020). Importantly, please note that BoV-AE is also useful with larger vectors: On Yelp-Reviews, which are substantially longer than Yelp-Sentences, even 512 dimensions are not enough for the fixed AE to encode the text well (Figure 7, appendix). Consequently, using BoV-AEs in this case is beneficial. It is likely that this effect would be even larger on texts that are even longer than in Yelp-Reviews. In summary, our method attempts to fix a fundamental shortcoming of Emb2Emb with fixed-sized autoencoders.
> > > * __“Why the fixed AE uses the average of representations of encoder, not the decoder with only one step autoregressive decoding? The number of parameters may affect the performance as well.”__
> > > If we understand correctly, here you are referring to the encoder-decoder architecture used during autoencoder pretraining. Here we keep as many components fixed to enable controlled experiments. Fixed AE and BoV-AE both use a standard Transformer encoder to produce a set of vectors $\mathbb{X} = ${$ \mathbf{z}_1, …, \mathbf{z}_n $}. Then there is a compression step, say $\mathbb{X’} = \operatorname{compress(\mathbb{X})}$, which reduces the number of vectors in the latent representation. This is the only component that differs between fixed AE and BoV-AE (described below). Finally, a standard Transformer decoder reproduces the input text given the (compressed) representation $\mathbb{X’}$. Concretely, in BoV-AE, compression is done through L0Drop. In the fixed AE model, the compression consists of mean pooling over the representation. This is the standard practice for obtaining (single-vector) sentence embeddings of texts, e.g., in Sentence-BERT [1]. It is also what is used in Mai et al. (2020). So we don’t see a reason for using a different way of obtaining a single-vector representation.
> > > Specifically, it is not quite clear to us what you mean by “the decoder with only one step autoregressive decoding”. Do you suggest to use a Transformer encoder-decoder model, and use the decoder autoregressively for one step?
> > > * __“the mapping seems not to be a real mapping between bag of vectors since the latter vectors depend on former vectors”__ It is true that the latter vectors depend on the former vectors during generation, but note that our loss functions are defined on bags-of-vectors, not on sequences of vectors: We DO NOT provide gold labels at every time step of the autoregressive predictions. Rather, we propose a differentiable variant of the modified Hausdorff distance loss, which is specifically designed for sets, NOT sequences. That is, given a bag-of-vectors predicted by $\Phi$, the loss is indifferent with respect to the order in which the vectors were generated.
> > > * __“How about training a decoder with k learnable input vectors? In that way, you can decode the k bag-of-vectors in parallel so it should be real bag-of-vectors.”__
> > > We chose autoregressive generation of the output bag for simplicity, since it is known to work somewhat well even for generating sets of variable length. However, we are certainly open to suggestions to further improve this, so thank you. In the model you suggest, how would it be possible to generate bags of varying size? If k was chosen dynamically, how could we still ensure them to be learnable vectors?
> > >
> > > [1] Sentence-BERT: Sentence Embeddings using Siamese BERT-Networks
> > > https://arxiv.org/abs/1908.10084

---

> > > > ### Comment · Reviewer_jHbb · 2021-11-30
> > > > **Reply**
> > > >
> > > > Thanks for your addition response. My questions are resolved.

---

### Official Review · Reviewer_74E4 · 2021-11-09

**Correctness:** 3
**Technical Novelty And Significance:** 3
**Empirical Novelty And Significance:** 2
**Recommendation:** 5
**Confidence:** 4

**Main Review:**

I'm a bit confused how the mapping $\Phi$ is learned. As you generate vectors autoregressively, it's actually not a mapping between two bags of vectors, i.e., the probability of generating $(z_1,z_2)$ can be different from the probability of generating $(z_2,z_1)$. I also think there are many problems with the notations. In the text between Eq. (2) and Eq. (3), you said "$p_{gen}$ is a function of the output $z_t'$ of Transformer...". What is $z_t'$? Is it $\hat z_t$? But $\hat z_t$ is determined after $p_{gen}$. I couldn't follow this part. In Sec 4.2, what is $\mathbb Y$ in $\mathcal L(\mathbb X, \mathbb Y)$? I don't see it defined anywhere. Also, you've defined $\mathbb X:=\text{enc}(x)$ in the beginning of Sec. 3, but now you seem to use it as the output of $\Phi$.

The experimental results lag far behind previous work. On Yelp-Sentences, He et al. (2020) achieved accuracy 87.90 and self-BLEU 48.38, while in Fig. 5 the proposed model has self-BLEU only 20 for the same level of accuracy. On sentence summarization, the proposed model has ROUGE-L only 18.3, while Rush et al. reached 28.34 in 2015, not to mention the results in recent years. Although it's not completely comparable as you use less training data and no parallel sentences, it is unclear whether the proposed model is usable with such poor results. There are no sentence examples provided, nor is human evaluation conducted.

[1] He et al. (2020). A Probabilistic Formulation of Unsupervised Text Style Transfer.


**Summary Of The Paper:**

This paper proposes a text autoencoder with a bag-of-vectors embedding, which is more capable of encoding longer text than a single-vector embedding. To train such a model, the authors first encode the input sequence into a vector sequence of equal length, and then add a gate to each vector and encourage the model to keep a small number of open gates.

The proposed BoV-AE can be used in (non-parallel) seq2seq tasks by learning a mapping within the embedding space, similar to the Emb2Emb method proposed by Mai et al. The difference is that the mapping is now between two bags of vectors instead of between two vectors. The authors use a Transformer with a copy mechanism and an offset vector to produce a bag of vectors. To align two bags of vectors, the authors adjust the average Hausdorff distance to a soft version to make training smoother.

Experiments are conducted on a sentiment transfer task and a sentence summarization task. BoW-AE shows better performance compared to AE with a single-vector embedding.



**Summary Of The Review:**

I recommend rejecting this paper because of its unclear method description, poor results, and insufficient evaluations.

---

> ### Author Response · Authors · 2021-11-21
> **Reply to review**
>
> Thank you for your review.
> * __“As you generate vectors autoregressively, it’s actually not a mapping between two bags of vectors.”__: Indeed, we use an autoregressive prediction model for $\Phi$ for simplicity. But note that our loss functions are defined on bags-of-vectors, not on sequences of vectors: We DO NOT provide gold labels at every time step of the autoregressive predictions. Rather, we propose a differentiable variant of the modified Hausdorff distance loss, which is specifically designed for sets, NOT sequences. That is, given a bag-of-vectors predicted by $\Phi$, the loss is indifferent with respect to the order in which the vectors were generated.
> * __“I also think there are many problems with the notations.”__: We updated our notations in many parts of the paper to make it clearer. Specifically, we introduce new variables in the two cases you mentioned to make it easier to follow.
> * __“The experimental results lag far behind previous work. [...] it is unclear whether the proposed model is usable with such poor results.”__:
> We also address this point in our reply to all reviewers. We are not proposing a usable system, but conducting controlled experiments on the usefulness of our specific contributions.  Our goal is to develop a method that allows learning in the embedding space of autoencoders with variable-size latent representations, an issue that is at the core of the representation learning community. We formulate clear hypotheses about the benefits of our model and provide ablations that show that our technical contributions are necessary. We believe that our paper should be evaluated on its scientific merits, not on whether it tops a leaderboard or not. More specifically, we compare our autoencoders with variable-size latent representations to comparable models with fixed-sized latent representations, essentially the Emb2Emb models of Mai et al. (2020). These previous Emb2Emb models with fixed-sized vectors achieved comparable and better results than a related state-of-the-art model at the time [1].  It is true that He et al. (2020) achieve much better scores with the addition of pretrained language models, but that is an orthogonal issue, which would make the experiments intractable for us to conduct.
>
> [1] Controllable Unsupervised Text Attribute Transfer via Editing Entangled Latent Representation - Ke Wang, Hang Hua, Xiaojun Wan
> https://arxiv.org/abs/1905.12926

---

### Author Response · Authors · 2021-11-21
**Reply to all and changes to the paper**

We thank all reviewers for their feedback. Here, we would like to address criticism that is shared by multiple reviewers. These are:

* Low scores in comparison to other published work
* No comparison to systems outside of the Emb2Emb framework

For the tasks we use for evaluation, recent work has reported better results using large pretrained language models. We cannot use these models directly in our studies because they are not autoencoders, and we do not have the resources necessary to pretrain our own large-scale autoencoders.  We cannot fairly compare previous results using these large pretrained models to our results without them, so if the reviewers required competitive results with the state of the art in these tasks, then the kind of work we are reporting in this paper would be impossible.
However, the immense success of pretrained language models reinforces the relevance of the Emb2Emb framework to current research, given that it is based on pretrained autoencoders, with a plug-and-play approach.  We therefore use the methodology of controlled experiments testing specific hypotheses, rather than chasing leaderboards.  Many researchers value this approach to science (e.g. [1]), and thus we do not accept these as valid criticisms of our work.

[1] Efficient NLP - Yuki Arase, Phil Blunsom, Mona Diab, Jesse Dodge, Iryna Gurevych, Percy Liang, Colin Raffel, Andres Rueckle, Roy Schwartz, Roy Schwartz, Noah A. Smith, Emma Strubell, Yue Zhang. - https://public.ukp.informatik.tu-darmstadt.de/enlp/Efficient-NLP-policy-document.pdf

__Changes to the paper draft__:
We made the following changes to the paper:
* to clarify Figure 3 and 5, we added a half sentence to the caption of Fig 5 and to the end of the first paragraph of Section 5.1.
* fixed the typo in Equation 2: $X$ -> $\mathbb{X}$
* fixed the typo in Figure 2 caption: $\mathbf{\hat{z}}_x$ -> $\mathbf{\hat{z}}_y$
* for illustration, we added an example of the loss to the first paragraph of 4
* fixed the typo in the reference to the full graph in the first paragraph of 5.1: Figure 8 -> Figure 7
* added an explicit description of the length loss in first paragraph of 5.3
* in Section 4, we explicitly introduce the notation for the input bag $\mathbb{X}$ and the predicted output bag $ \hat{\mathbb{X}} $ and use them across the section.
* we added a section on the computational complexity of BoV-AEs in Appendix B.2.2.
* we added a qualitative analysis of outputs on Yelp-Sentences in Appendix B.2.3.

---

### Author Response · Authors · 2021-11-29
**Note on Autoregressive Mapping $\Phi$**

Our mapping $\Phi$ generates the output bag autoregressively. Multiple reviewers have raised the concern that this implies that the output of the mapping cannot actually be a set. We would like to point out that this is a misconception. Whether an object is a set or a sequence does not depend on how it is *generated*, but rather how it is *interpreted*. If it is interpreted in a way that order does not matter, it is not a sequence. The output of the mapping is interpreted in two places: a) In the loss that is used for training the mapping and b) in the Transformer decoder. Both of these interpret the output as a set. In b), cross-attention (without positional embeddings) is used to access the vectors in the bag. We explain in Section 3 that attention is a function defined on sets. In a), the loss consists of two terms: The (differentiable) Hausdorff distance, which is a set loss (see Section 4.3), and the style classifier loss, which is a Transformer encoder (without positional embeddings), which again uses attention.

---

### Decision · Program_Chairs · 2022-01-20

**Decision:**

Reject

**Comment:**

The paper addresses unsupervised conditional text generation extending emb2emb (Mai et al, 2020) with bag-of-vectors antoencoders.

Reviewers shared several concerns about the clarity of this paper and empirical results.